# Health systems performance in health outcomes, health financing and COVID-19 pandemic: Lessons from 31 countries

Pirhossein Kolivand[1,2‡], Jalal Arabloo[3‡], Peyman Saberian[1,4], Taher Dorooudi[1], Soheila Rajaie[1], Fereshte Karimi[1], Behzad Raei[5], Masoud Behzadifar[6], Arash Parvari[7], Seyed Jafar Ehsanzadeh[8], Saeid Homayoun[9], Shahrzad Salehbeigi[10], Peyman Namdar[11], Samad Azari [iD][3]*

**1** Research Center for Emergency and Disaster Resilience, Red Crescent Society of the Islamic Republic of Iran, Tehran, Iran, **2** Department of Health Economics, Faculty of Medicine, Shahed University, Tehran, Iran, **3** Health Management and Economics Research Center, Iran University of Medical Sciences, Tehran, Iran, **4** Department of Anesthesiology, Imam Khomeini Hospital Complex, Tehran University of Medical Sciences, Tehran, Iran, **5** Department of Health, Safety, and Environment Management, School of Public Health, Zanjan University of Medical Sciences, Zanjan, Iran, **6** Social Determinants of Health Research Center, Lorestan University of Medical Sciences, Khorramabad, Iran, **7** Department of Epidemiology and Biostatistics, School of Public Health, Tehran University of Medical Sciences, Tehran, Iran, **8** English Language Department, School of Health Management and Information Sciences, Iran University of Medical Sciences, Tehran, Iran, **9** Faculty of Education and Economics, University of Gävle, Gävle, Sweden, **10** Department of Cardiology, ST. Antonius Hospital, Cologne, Germany, **11** Department of Emergency Medicine, Metabolic Disease Research Center, Qazvin University of Medical Sciences, Qazvin, Iran

‡ Contributed equally as the first authors.
* samadazari1010@gmail.com

## Abstract

### Background

ssssHealth system performance is a multifaceted concept that encompasses various dimensions of a nation's healthcare infrastructure. This study aims to assess and rank the performance of health systems across different regions of the world.

### Methodology

We employed the Technique for Order of Preference by Similarity to Ideal Solution (TOPSIS) method in 2023 to evaluate and rank the health system performance of 31 countries across six geographical regions. Our evaluation included six general categories and twelve indicators related to health, finance, and the COVID-19 pandemic. The final weights for these indicators were determined using the Three-scale method and the Entropy-weighting method. Additionally, we categorized health system performance into three groups: high, moderate, and low. Hierarchical clustering of health system performance scores was conducted using SPSS software (version 26).

**Data availability statement:** All data underlying the findings described in their manuscript are freely available as an appendix.

**Funding:** The author(s) received no specific funding for this work.

**Competing interests:** The authors have no relevant financial or non-financial interests to disclose.

## Results

Luxembourg emerged as the only high-performing health system, while Qatar and the Netherlands fell into the moderate-performance group. Other countries exhibited low-performing health systems. Notably, within the low-performance group, the United States of America, Australia, Singapore, Canada, England, and Germany achieved relatively better rankings. Conversely, Yemen, Egypt, Afghanistan, and Bolivia ranked lowest in terms of health system performance.

## Conclusion

Contrary to the assumption that higher health spending guarantees improved performance, the experience of COVID-19 among high-income countries revealed mixed results. Strengthening resilience, investing in public health systems, and ensuring sustainable financial resources are crucial for enhancing health system performance.

## 1. Background

The healthcare system encompasses all organizations, institutions, resources, and personnel working toward the common goal of improving societal health. These activities involve addressing factors that influence health and directly enhancing the well-being of the population [1]. Health system frameworks serve as conceptual tools that delineate and describe the objectives of the health system, as well as the factors influencing its performance [2].

Health Systems Performance (HSP) involves a comprehensive assessment and quantification of a health system's effectiveness, efficiency, equity, and quality in achieving its goals. This evaluation entails continuous monitoring, analysis, and dissemination of data across various dimensions of the health system, including governance, financial management, resource allocation, service delivery, and health outcomes. The primary purpose is to provide evidence-driven insights to policymakers, enabling informed decision-making, prioritization of interventions, and optimization of overall health system functionality to address the dynamic challenges of contemporary healthcare. The evaluation framework must recognize the interconnectedness of diverse health system domains and objectives, emphasizing inputs, processes, outputs, outcomes, and impacts to ensure transparency, accountability, and ongoing improvements in healthcare delivery and public health outcomes [3].

Numerous frameworks have been proposed to measure HSP. These frameworks consider various indicators, including health goals, health inequalities, coverage, sustainable and equitable financing, quality, patient satisfaction, allocation efficiency, technical efficiency, cost-effectiveness, political acceptability, and financing mechanisms [4–6]. The World Health Organization (WHO) report in 2000 distinguishes between final goals (such as responsiveness and fair financing) and intermediate goals within health systems. It emphasizes that intermediate goals play a pivotal role in achieving the ultimate objectives. Accurate measurement of goals and HSP,

informed decision-making processes, and policy adoption are essential. For instance, one criterion for assessing performance is the level of coverage achieved [7].

At the national level, healthcare systems exhibit significant variations in outcomes, even among countries with similar income levels. Some of these disparities can be attributed to differences in HSP. Consequently, healthcare policymakers should assess HSP at all levels, identify factors influencing access to and utilization of health services, and formulate policies to enhance population health. However, it is essential to recognize that improvements in availability and access, especially in resource-constrained settings, do not always directly translate into better health outcomes [8].

Strengthening healthcare systems necessitates developing management capacity across all levels. Key competencies for healthcare system management include strategic thinking, human resource management, financial stewardship, performance evaluation, governance, leadership, and community engagement [9].

Numerous countries have established mechanisms to evaluate the performance of their health systems. In the United States, the National Scorecard by the Commonwealth Fund assesses health system performance (HSP) across domains such as health outcomes, quality, and equity [10]. Belgium conducts a periodic (HSPA) using 80 indicators. England closely monitors the National Health Service (NHS) through an extensive range of indicators [11]. Meanwhile, the Netherlands publishes a biennial Health Care Performance Report, focusing on quality, access, and cost considerations [12]. In Africa, South Africa's Health Systems Trust releases a district health barometer [13], while Uganda and Ghana employ various tools, including league tables and comprehensive assessments, to evaluate HSP. Despite extensive research on Health System Performance Assessments (HSPA) in high-income countries, low-income countries have fewer documented experiences, underscoring the need for further investigation in this area [14].

In Iran, HSP has undergone evaluation. The results indicate that Iran's health system has made progress in achieving universal health coverage (UHC). However, challenges persist in resource utilization, necessitating opportunities for enhancing technical efficiency. Studies emphasize that improving Iran's health system performance requires better implementation, resource provision, facility management, and effective governance [15].

Recent years have witnessed the profound impact of the COVID-19 pandemic on health systems worldwide. This global crisis has rigorously tested the resilience and performance of health systems, prompting the World Health Organization (WHO) to underscore the importance of assessing its impact on HSP using key indicators. Research highlights the imperative for health systems to enhance preparedness and response capabilities to effectively manage future public health emergencies. Furthermore, the pandemic underscores the significance of universal health coverage, robust public health infrastructure, and intergovernmental coordination in optimizing HSP during crises. Understanding and measuring the effects of pandemics on health systems are critical for informed policy decisions and strengthening health systems to better respond to future public health challenges, drawing insights from successful practices in high-performing countries [16–18].

Given this, the aim of this study is to assess health system performance in selected countries and compare it with Iran. The findings will provide comprehensive information based on healthcare indicators, informing policy-making in this sector.

## 2. Methods

### 2.1. Study design

The Technique for Order Preference by Similarity to an Ideal Solution (TOPSIS) method, developed by Hwang and Yoon in 1981, employs the concept of positive and negative ideal solutions as benchmarks for evaluating and ranking performance across units of analysis [19]. This method assesses the strengths and weaknesses of existing objects by ranking them based on their proximity to these idealized targets. The ranking process involves measuring the distance between each evaluation object and both the best and worst solutions [20,21]. The TOPSIS method can be summarized in the following steps [22]:

1. **Calculate the Decision Matrix after Normalization.**

2. **Calculate the Decision Matrix after Normalization and Weighting.**

3. **Determine the Positive-Ideal and Negative-Ideal Solutions.**

4. **Calculate Separation Measures using n-Element Euclidean Distance.**

5. **Calculate the Relative Closeness to the Ideal Solution**, which ranges from zero to one.

6. **Rank the Preference Order**.

The TOPSIS method is highly objective, eliminating the influence of subjective factors and maximizing the utilization of original data with minimal loss of information [23]. Consequently, it has found effective application in various research domains, including supply chain management, logistics, engineering, and business systems. Notably, it is well-suited for risk assessment and evaluating system performance [24].

## 2.2. Study area and data source

This study employs the TOPSIS method to evaluate and rank health system performance (HSP) across six global regions: Africa, North America, South America, Europe, the Middle East, and Asia Pacific [25]. The study aims to compare health system performance. Due to practical constraints, it was not feasible to include every country from the analyzed regions. Therefore, a purposive sampling approach was adopted, resulting in the selection of 31 countries. These countries were chosen based on criteria such as GDP level, population size, COVID-19 indices, and healthcare expenditure. Specifically, countries with low, middle, and high GDP levels, along with other relevant indicators, were included from each region (see Fig 1). The HSP in these selected countries was evaluated and ranked using desired performance indicators. Data were sourced from the World Health Organization (WHO) database and the Institute for Health Metrics and Evaluation (IHME), which provides data and resources from the Global Burden of Disease Study (GBD). All data is free and accessible.

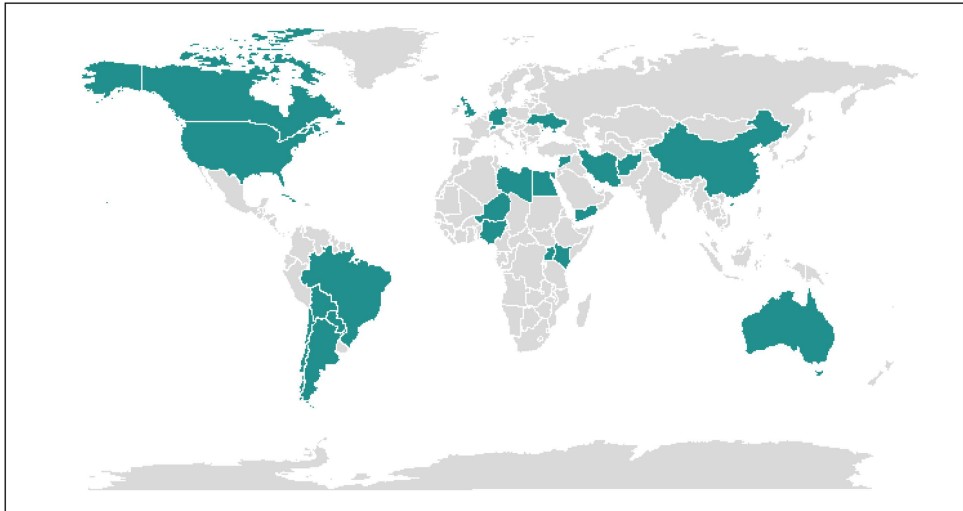

**Fig 1. Countries included in this study.** The current map includes the selected countries based on the indicators of GDP, population, cases of the COVID-19 and health expenditure of a range of low, moderate and high countries. The 6 regions included Africa, North America, South America, Europe, the Middle East, and Asia Pacific.

### 2.3. Calculation methodology

In this study, we delineate the following components of our research methodology:

1. **Framework Design:**

   ◦ We initially crafted a comprehensive framework for evaluating health system performance (HSP) based on three critical dimensions: health indicators, financial indicators, and COVID-19 impact. Drawing upon existing frameworks, including the World Health Organization's six building blocks [2,4,26], we aimed to create a robust assessment tool.

   ◦ **Health Indicators:** Our selected health indicators include Neonatal Mortality Rate (NMR), Maternal Mortality Ratio (MMR), Healthcare Access and Quality Index (HAQ), Health-Adjusted Life Expectancy (HALE), and Non-Communicable Disease (NCD) prevalence.

   ◦ **Financial Indicators:** We considered Out-of-Pocket Spending, Prepaid Private Spending, Government Spending, and GDP per Capita as essential financial metrics.

   ◦ **COVID-19 Indicators:** To account for the pandemic's impact, we incorporated Total Cases, Total Deaths, and Total Vaccinations related to COVID-19.

2. **Weight Assignment:**

   ◦ Next, we assigned weights to each indicator using a combination of established weighting methods. These weights reflect the relative importance of each dimension in assessing HSP.

3. **Composite Index Calculation:**

   ◦ Leveraging available data, we computed a composite index that allows us to measure HSP across selected countries globally. Our primary objective is to rank these countries using the Technique for Order Preference by Similarity to an Ideal Solution (TOPSIS) method.

4. **Hierarchical Cluster Analysis:**

   ◦ We employed Hierarchical Cluster Analysis to group countries based on their performance in the three dimensions: COVID-19 impact, financing, and health system.

5. **Country Clustering:**

   ◦ Using the Complete Linkage method, we clustered countries into three classes: high, moderate, and low performers

The Fig 2 shows the selected indicators to evaluate the HSP. By reviewing the existing literature and performance frameworks, the researchers proposed a framework to compare the performance of countries. In this framework, various indicators were selected in the three areas of financing, health and indicators related to COVID-19.

**2.3.1. The HSP framework.** Assessing health system performance (HSP) through a combination of quantitative and qualitative methods represents an effective approach to fortifying national health systems. This comprehensive evaluation provides a holistic understanding of system strengths and weaknesses, empowering policymakers to make informed decisions for enhancement [13,27]. By scrutinizing the World Health Organization's (WHO) HSP framework [4] and integrating insights from relevant studies and the objectives of our study, we constructed an evaluation index framework for quantitative analysis of HSP (see Fig 2).

**2.3.2. Index weighting.** In this study, we employed a hybrid approach, combining the Three-Scale method and the Entropy-Weighting method, to determine the final weights for the indicators. This approach ensures a robust and objective

assessment of each indicator's importance in the overall evaluation of the studied topic. The Entropy-Weighting method yields estimates comparable to propensity score methods, enhancing the reliability of our weight assignments [28]. Additionally, the Three-Scale method offers practical advantages, facilitating expert judgments, constructing a priority judgment matrix, and logically ranking priorities [29].

**2.3.3. Classification of HSP.** To cluster countries based on their health system performance, we utilized version 26.0 of the SPSS software. Hierarchical clustering was applied to HSP scores for 31 countries across six global regions. To determine the optimal number of clusters, we employed the Silhouette method. Our analysis revealed that three clusters best represent the data (see Fig 3). Employing the Complete Linkage method, we categorized countries into three classes: high, moderate, and low performers. The Complete Linkage method, a hierarchical clustering technique, considers the maximum distance between any two points in different clusters, facilitating the identification of distinct groups based on the similarity of HSP scores among countries.

| Health System Performance | | |
|---|---|---|
| **Financial Indicators** | **Health Indicators** | **Covid-19 Indicators** |
| Out of pocket spending as % of total health spending | Neonatal mortality rate (per 1000 live births) Both sexes | Total cases |
| Prepaid private spending as % of total health spending | Maternal mortality ratio (per 100 000 live births) | Total deaths |
| Government spending as % of total health spending | HAQ index | Total vaccinations |
| GDP per Capita (U$) | HALE index | |
| | NCD | |

**Fig 2. Indicators included in this study (HSP Framework).**

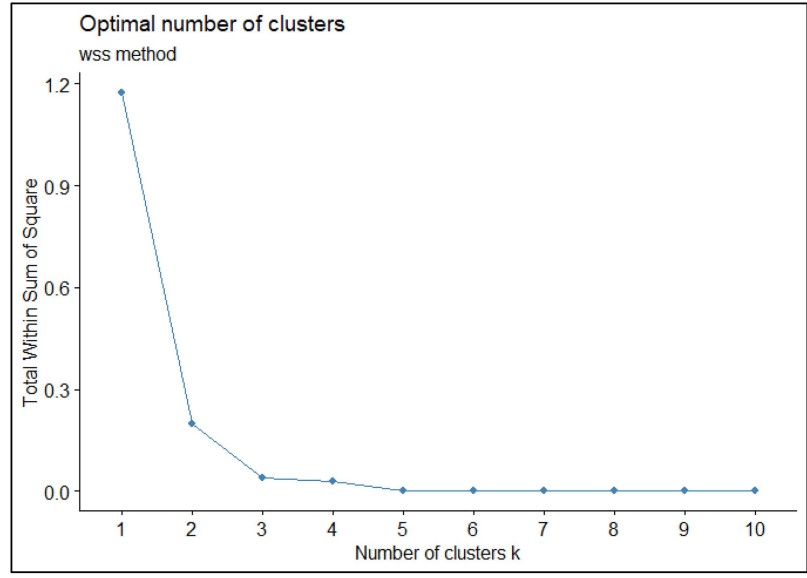

**Fig 3. Optimal number of clusters.**

## 2.4. Ranking countries based on indicators

In addition to employing the TOPSIS method, we conducted a comparative analysis of 31 countries included in the study. These countries were ranked based on health, financial, and COVID-19 indicators for the years 2000, 2019, 2020, and 2023 (for COVID-19). The objective was to provide a comprehensive overview of each country's performance relative to the selected indicators.

## 2.5. Ethical approval

This study is an excerpt from the research project with the Ethical Code IR.RCS.REC.1401.013 from the Iranian Red Crescent Society, conducted at the Research Center for Emergency and Disaster Resilience, Red Crescent Society of the Islamic Republic of Iran, Tehran, Iran.

## 3. Results

### 3.1. Clustering of countries

To analyze and discuss health system performance, we achieved optimal results by clustering the 31 countries into three distinct groups (K = 3). Fig 4 illustrates the cluster chart generated through hierarchical cluster analysis, visually representing the clustering outcomes. Additionally, Table 1 presents the division of countries into three categories: high, moderate, and low performance, facilitating a clear understanding of each country's health system performance classification.

Furthermore, we created a global map (Fig 5) displaying health system performance values. This visual representation highlights variations across different regions and countries, aiding in the identification of areas that require targeted interventions and improvements.

Table 1 shows the ranking results of the countries in the order of high, Moderate and low performance.

In our analysis, the majority of countries were classified in the category of low performance. Using the method of clustering countries based on their performance, one country (Luxembourg) had high performance, while two countries (Switzerland and Qatar) had moderate performance. On the other hand, 28 countries, including Iran, had low performance.

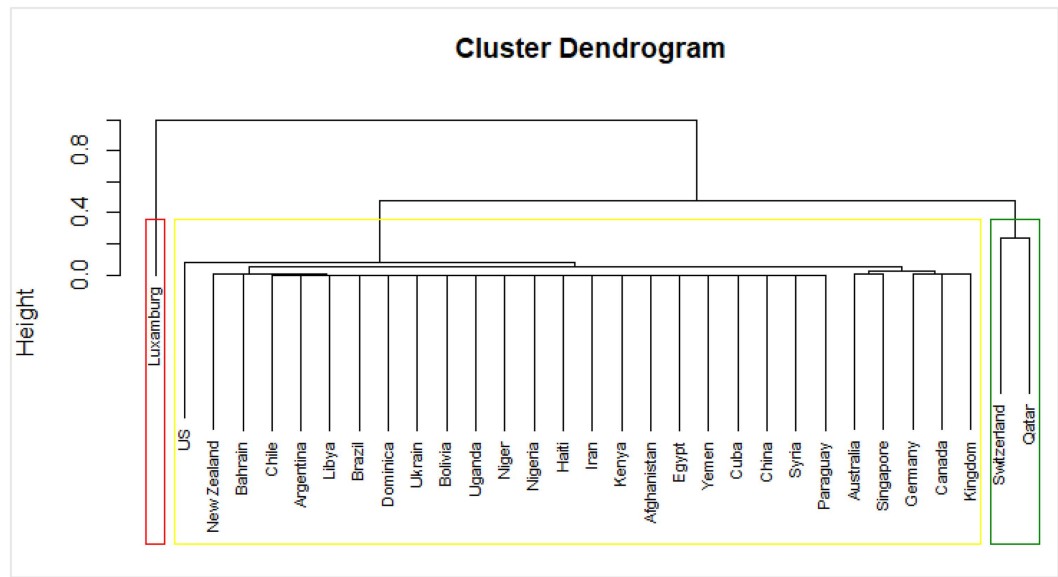

**Fig 4. Clustering spectrum diagram.**

**Table 1. Health system performance by the clustering countries.**

| Level of performance | Country |
|---|---|
| High performance [1] | Luxembourg |
| Moderate performance [2] | Switzerland, Qatar |
| Low performance [27] | United States, Australia, Singapore, Canada, United Kingdom, Germany, New Zealand, Bahrain, Chile, Argentina, Libya, Brazil, Dominica, Cuba, China, Syria, Paraguay, Iran, Kenya, Nigeria, Haiti, Uganda, Niger, Ukraine, Bolivia, Afghanistan, Egypt, Yemen |

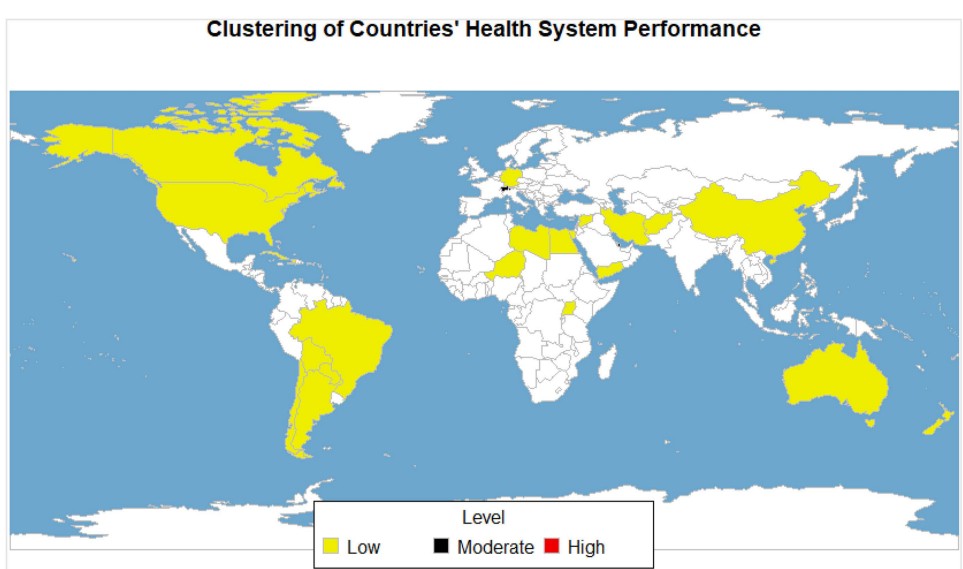

**Fig 5. Global distribution of HSP assessment and ranking (31 countries).**

### 3.2. Ranking countries with TOPSIS method

Table 2 presents the ranking of the HSP of the 31 countries. The three countries, Luxembourg, Switzerland, and Qatar, are ranked in the top three among 31 countries and have shown better performance in terms of health, financing, and COVID-19 indicators. In this ranking, Iran is ranked 21st. On the other hand, the three countries of Afghanistan, Egypt, and Yemen occupied the bottom three ranks and did not perform well (Table 2).

### 3.3. Ranking countries with the value of indicators

**3.3.1. Health indicators.** Table 3 shows the ranking of the countries for the MMR and NMR indicators. These rankings are arranged for the years 2000 and 2019. Also, the change in the rank of each country in the form of red and green arrows has shown the deterioration or improvement of the situation of each country. For example, the Australia had the best status in the MMR index in 2000, while in 2019, it had a decrease in this index. While the rank of Egypt improved and moved from 21st to 13th. For the index NMR the Singapore has been the best in both years. While the Afghanistan is still at the bottom of the ranking table (Table 3).

**Table 2. Global health system performance rankings of the 31 countries.**

| Country | $d_i^+$ | $d_i^-$ | $C_i$ | Rank |
|---|---|---|---|---|
| Luxembourg | 48.1483104 | 16707313689 | 0.999999997 | 1 |
| Switzerland | 4369573610 | 3988440514 | 0.477199542 | 2 |
| Qatar | 6832008125 | 2171637251 | 0.241195334 | 3 |
| USA | 9924129761 | 878355219.9 | 0.081310478 | 4 |
| Australia | 11048581714 | 582973400.3 | 0.050119988 | 5 |
| Singapore | 11263870811 | 534799289.4 | 0.045327082 | 6 |
| Canada | 12104544205 | 370043484.8 | 0.029663785 | 7 |
| UK | 12237841114 | 347167248.2 | 0.027585778 | 8 |
| Germany | 12454021060 | 311862421.4 | 0.024429365 | 9 |
| New Zealand | 13862583362 | 132662990.4 | 0.009479146 | 10 |
| Bahrain | 15660989064 | 16930915.41 | 0.001079921 | 11 |
| Chile | 16451286490 | 1005733.284 | 6.11303E-05 | 12 |
| Argentina | 16540292813 | 436732.7841 | 2.64035E-05 | 13 |
| Libya | 16539432917 | 430946.7476 | 2.6055E-05 | 14 |
| Brazil | 16591996566 | 212457.2066 | 1.28046E-05 | 15 |
| Dominica | 16630271347 | 104429.0067 | 6.27941E-06 | 16 |
| Cuba | 16651021008 | 64041.31437 | 3.84607E-06 | 17 |
| China | 16666621492 | 40865.30286 | 2.45192E-06 | 18 |
| Syria | 16676470021 | 31104.73837 | 1.86518E-06 | 19 |
| Paraguay | 16681459698 | 25923.9707 | 1.55406E-06 | 20 |
| Iran | 16676565606 | 18470.92694 | 1.1076E-06 | 21 |
| Kenya | 16706567733 | 17529.04835 | 1.04923E-06 | 22 |
| Nigeria | 16703908692 | 16995.05647 | 1.01743E-06 | 23 |
| Haiti | 16706528560 | 16508.1971 | 9.88128E-07 | 24 |
| Uganda | 16707214954 | 15116.49784 | 9.04788E-07 | 25 |
| Niger | 16707296830 | 14803.22813 | 8.86033E-07 | 26 |
| Ukraine | 16699439966 | 13837.09784 | 8.28596E-07 | 27 |
| Bolivia | 16702917675 | 13817.08125 | 8.27225E-07 | 28 |
| Afghanistan | 16707289983 | 8450.919657 | 5.05822E-07 | 29 |
| Egypt | 16701989641 | 4567.737292 | 2.73485E-07 | 30 |
| Yemen | 16706842863 | 1004.62683 | 6.01327E-08 | 31 |

$d_i^+$ and $d_i^-$ represents the distance between the weighted value of each option to the positive ideal alternative and the negative ideal alternative respectively. $C_i$ represents the similarity index. It indicates the relative closeness to the ideal alternative and is used to prioritize the countries.

Table 4 shows the ranking of the HALE and the Non-Communicable Diseases rate indicators. Iran has improved the most in the HALE index, its rank has changed from 17 to 12. In the Non-Communicable Diseases rate index, Afghanistan's rank has improved from 20 to 9 and has changed the most (Table 4).

**3.3.2. Financial indicators.** In Table 5, the ranking of financial indicators is shown. The status of the out-of-pocket spending indicator shows that the Libya and the Singapore have improved the most, while the Yemen and the Ukraine have had the biggest decline in the ranking. Also, the status of the index shows that Iran has improved the most and Syria has decreased the most. Switzerland and Cuba remain unchanged at the top and bottom of the table, respectively (Table 5).

The status of the Government spending and the GDP per capita indicators is shown in the Table 6. The Yemen dropped 11 ranks in the Government spending indicator. China also improved the most with a change of 13 ranks. GDP per

**Table 3. Ranking of 31 countries (MMR & NMR Indices).**

| Maternal Mortality Rate (per 100 000 live births) | | | | Neonatal Mortality Rate (per 100 000 live births) | | | |
|---|---|---|---|---|---|---|---|
| Country | 2000 Rank | 2019 Rank | Change in rank | Country | 2000 Rank | 2019 Rank | Change in rank |
| Australia | 1 | 2 | ↓ - 1 | Singapore | 1 | 1 | 0 |
| Germany | 2 | 1 | ↑ + 1 | Luxemburg | 2 | 2 | 0 |
| Switzerland | 3 | 6 | ↓ - 3 | Germany | 3 | 3 | 0 |
| Luxemburg | 4 | 4 | 0 | Switzerland | 4 | 7 | ↓ - 3 |
| Canada | 5 | 10 | ↓ - 5 | New Zealand | 5 | 6 | ↓ - 1 |
| New Zealand | 6 | 7 | ↓ - 1 | Australia | 6 | 5 | ↑ + 1 |
| UK | 7 | 8 | ↓ - 1 | Canada | 7 | 10 | ↓ - 3 |
| USA | 8 | 15 | ↓ - 7 | UK | 8 | 8 | 0 |
| Singapore | 9 | 3 | ↑ + 6 | Cuba | 9 | 4 | ↑ + 5 |
| Bahrain | 10 | 11 | ↓ - 1 | Bahrain | 10 | 9 | ↑ + 1 |
| Qatar | 11 | 5 | ↑ + 6 | USA | 11 | 12 | ↓ - 1 |
| Chile | 12 | 12 | 0 | Libya | 12 | 14 | ↓ - 2 |
| Syria | 13 | 17 | ↓ - 4 | Chile | 13 | 15 | ↓ - 2 |
| Ukraine | 14 | 9 | ↑ + 5 | Qatar | 14 | 11 | ↑ + 3 |
| Iran | 15 | 16 | ↓ - 1 | Ukraine | 15 | 16 | ↓ - 1 |
| Cuba | 16 | 19 | ↓ - 3 | Argentina | 16 | 17 | ↓ - 1 |
| Libya | 17 | 22 | ↓ - 5 | Syria | 17 | 22 | ↓ - 5 |
| China | 18 | 14 | ↑ + 4 | Dominica | 18 | 27 | ↓ - 9 |
| Brazil | 19 | 20 | ↓ - 1 | Paraguay | 19 | 21 | ↓ - 3 |
| Argentina | 20 | 18 | ↑ + 2 | Iran | 20 | 18 | ↑ + 2 |
| Egypt | 21 | 13 | ↑ + 8 | Brazil | 21 | 19 | ↑ + 2 |
| Dominica | 22 | 23 | ↓ - 1 | China | 22 | 13 | ↑ + 9 |
| Paraguay | 23 | 21 | ↑ + 2 | Egypt | 23 | 20 | ↓ - 3 |
| Yemen | 24 | 25 | ↓ - 1 | Kenya | 24 | 24 | 0 |
| Bolivia | 25 | 24 | ↑ +1 | Bolivia | 25 | 23 | ↑ + 2 |
| Haiti | 26 | 27 | ↓ - 1 | Haiti | 26 | 26 | 0 |
| Uganda | 27 | 26 | ↑ + 1 | Uganda | 27 | 25 | ↑ + 2 |
| Kenya | 28 | 29 | ↓ - 1 | Yemen | 28 | 28 | 0 |
| Niger | 29 | 28 | ↓ - 2 | Niger | 29 | 29 | 0 |
| Nigeria | 30 | 31 | ↓ - 1 | Nigeria | 30 | 30 | 0 |
| Afghanistan | 31 | 30 | ↑ + 1 | Afghanistan | 31 | 31 | 0 |

capita indicator had the most improvement and 10 rank changes in the years 2000–2019. The Syria also had the biggest decrease with 15 degrees (Table 6).

**3.3.3. COVID-19 indicators.** Given the recent emergence of COVID-19, this section covers a distinct time frame. We present rankings for two key indicators: the number of deaths and vaccinations (see Table 7).

1. **Number of Deaths:**

   ◦ We investigated deaths from 2020 to 2023.

   ◦ Notably, the United States consistently ranked at the bottom during this period.

   ◦ Luxembourg demonstrated significant improvement, moving from 10th to 5th place.

2. **Vaccination Indicator:**

   ◦ In terms of vaccinations, China, the United States, Brazil, and Germany maintained their top positions from 2021 to 2023.

   ◦ Conversely, Yemen, Haiti, and Dominica remained at the bottom of the list.

**Table 4. Ranking of 31 countries (HALE & NCDs Indices).**

| | HALE Index | | | | NCD (Non-Communicable Diseases) Rate | | |
|---|---|---|---|---|---|---|---|
| Country | 2000 Rank | 2019 Rank | Change in rank | Country | 2000 Rank | 2019 Rank | Change in rank |
| Singapore | 1 | 1 | 0 | Qatar | 1 | 1 | 0 |
| Canada | 2 | 4 | ⬇ - 2 | Kenya | 2 | 5 | ⬇ - 3 |
| Switzerland | 3 | 2 | ⬆ + 1 | Uganda | 3 | 2 | ⬆ + 1 |
| Australia | 4 | 5 | ⬇ - 1 | Niger | 4 | 3 | ⬆ + 1 |
| Germany | 5 | 6 | ⬇ - 1 | Bahrain | 5 | 6 | ⬇ - 1 |
| Luxembourg | 6 | 3 | ⬆ + 3 | Libya | 6 | 10 | ⬇ - 4 |
| Cuba | 7 | 11 | ⬇ - 4 | Paraguay | 7 | 11 | ⬇ - 4 |
| New Zealand | 8 | 7 | ⬆ + 1 | Nigeria | 8 | 4 | ⬆ + 4 |
| UK | 9 | 9 | 0 | Singapore | 9 | 8 | ⬆ + 1 |
| Chile | 10 | 8 | ⬆ + 2 | Iran | 10 | 12 | ⬇ - 2 |
| Paraguay | 11 | 15 | ⬇ - 4 | Syria | 11 | 14 | ⬇ - 3 |
| U.S. | 12 | 17 | ⬇ - 5 | Yemen | 12 | 7 | ⬆ + 5 |
| Libya | 13 | 18 | ⬇ - 5 | Bolivia | 13 | 13 | 0 |
| Argentina | 14 | 13 | ⬆ + 1 | Brazil | 14 | 17 | ⬇ - 3 |
| China | 15 | 10 | ⬆ + 5 | Chile | 15 | 18 | ⬇ - 3 |
| Dominica | 16 | 22 | ⬇ - 6 | Egypt | 16 | 16 | 0 |
| Syria | 17 | 21 | ⬇ - 4 | Haiti | 17 | 15 | ⬆ + 2 |
| Bahrain | 18 | 14 | ⬆ + 4 | China | 18 | 22 | ⬇ - 4 |
| Iran | 19 | 12 | ⬆ + 7 | Cuba | 19 | 26 | ⬇ - 7 |
| Qatar | 20 | 16 | ⬆ + 4 | Afghanistan | 20 | 9 | ⬆ + 11 |
| Brazil | 21 | 19 | ⬆ + 2 | Australia | 21 | 21 | 0 |
| Egypt | 22 | 24 | ⬇ - 2 | Argentina | 22 | 20 | ⬆ + 2 |
| Ukraine | 23 | 20 | ⬆ + 3 | Canada | 23 | 24 | ⬇ - 1 |
| Bolivia | 24 | 23 | ⬆ + 1 | New Zealand | 24 | 23 | ⬆ + 1 |
| Yemen | 25 | 25 | 0 | USA | 25 | 27 | ⬇ - 2 |
| Haiti | 26 | 29 | ⬇ - 3 | Switzerland | 26 | 25 | ⬆ + 1 |
| Kenya | 27 | 26 | ⬆ + 1 | Luxembourg | 27 | 19 | ⬆ + 8 |
| Nigeria | 28 | 28 | 0 | Dominica | 28 | 29 | ⬇ - 1 |
| Afghanistan | 29 | 31 | ⬇ - 2 | UK | 29 | 28 | ⬆ + 1 |
| Niger | 30 | 30 | 0 | Germany | 30 | 30 | 0 |
| Uganda | 31 | 27 | ⬆ + 4 | Ukraine | 31 | 31 | 0 |

For a visual representation of global health system performance, refer to the HSP map in Fig 5. This map highlights variations across regions and countries, aiding in identifying areas that require targeted interventions and improvements.

## 4. Discussion

### 4.1. The framework of health system performance (HSP)

The primary objective of our study is to evaluate Iran's health system performance in comparison to that of other selected countries. To achieve this, we employed a set of indicators related to health (Neonatal Mortality Rate, Maternal Mortality Ratio, Healthcare Access and Quality Index, Health-Adjusted Life Expectancy, and Non-Communicable Disease prevalence), COVID-19 (Total Cases, Total Deaths, and Total Vaccination), and financing (Out-of-Pocket Spending, Prepaid Private Spending, Government Spending, and GDP per Capita).

Several frameworks have been designed to assess HSP. Notably, the integrated healthcare performance framework proposes a range of indicators, including accountability, fair financial participation, efficiency, access, quality, value

**Table 5. Ranking of 31 countries (Out of Pocket Spending & Prepaid Private Spending Indices).**

| Out of pocket spending as % of total health spending | | | | Prepaid private spending as % of total health spending | | | |
|---|---|---|---|---|---|---|---|
| Country | 2000 Rank | 2019 Rank | Change in rank | Country | 2000 Rank | 2019 Rank | Change in rank |
| Germany | 1 | 6 | ↓ - 5 | Switzerland | 1 | 1 | 0 |
| Luxemburg | 2 | 1 | ↑ + 1 | USA | 2 | 2 | 0 |
| USA | 3 | 3 | 0 | Brazil | 3 | 3 | 0 |
| New Zealand | 4 | 5 | ↓ - 1 | Chile | 4 | 4 | 0 |
| Canada | 5 | 8 | ↓ - 3 | China | 5 | 17 | ↓ - 12 |
| Cuba | 6 | 2 | ↑ + 4 | Argentina | 6 | 14 | ↓ - 8 |
| Kingdom | 7 | 7 | 0 | Singapore | 7 | 5 | ↑ + 2 |
| Australia | 8 | 9 | ↓ - 1 | Kenya | 8 | 11 | ↓ - 3 |
| Bahrain | 9 | 15 | ↓ - 6 | Canada | 9 | 8 | ↑ + 1 |
| Argentina | 10 | 16 | ↓ - 6 | New Zealand | 10 | 10 | 0 |
| Qatar | 11 | 4 | ↑ + 7 | Paraguay | 11 | 12 | ↓ - 1 |
| Switzerland | 12 | 17 | ↓ - 5 | Germany | 12 | 16 | ↓ - 4 |
| Bolivia | 13 | 11 | ↑ + 2 | Qatar | 13 | 7 | ↑ + 6 |
| Dominica | 14 | 18 | ↓ - 4 | Bahrain | 14 | 15 | ↓ - 1 |
| Brazil | 15 | 13 | ↑ + 2 | Australia | 15 | 9 | ↑ + 6 |
| Chile | 16 | 20 | ↓ - 4 | Niger | 16 | 21 | ↓ - 5 |
| Uganda | 17 | 19 | ↓ - 2 | Kingdom | 17 | 23 | ↓ - 6 |
| Kenya | 18 | 14 | ↑ + 4 | Bolivia | 18 | 19 | ↓ - 1 |
| Ukraine | 19 | 26 | ↓ - 7 | Nigeria | 19 | 29 | ↓ - 10 |
| Yemen | 20 | 30 | ↓ - 10 | Ukraine | 20 | 24 | ↓ - 4 |
| Libya | 21 | 10 | ↑ + 11 | Haiti | 21 | 18 | ↑ + 3 |
| Singapore | 22 | 12 | ↑ + 10 | Iran | 22 | 6 | ↑ + 16 |
| Paraguay | 23 | 23 | 0 | Luxemburg | 23 | 22 | ↑ + 1 |
| Iran | 24 | 24 | 0 | Yemen | 24 | 28 | ↓ - 4 |
| Niger | 25 | 25 | 0 | Egypt | 25 | 13 | ↑ + 12 |
| China | 26 | 22 | ↑ + 4 | Libya | 26 | 25 | ↑ + 1 |
| Haiti | 27 | 27 | 0 | Syria | 27 | 26 | ↑ + 1 |
| Syria | 28 | 21 | ↑ + 7 | Uganda | 28 | 20 | ↑ + 8 |
| Egypt | 29 | 28 | ↑ + 1 | Afghanistan | 29 | 30 | ↓ - 1 |
| Nigeria | 30 | 29 | ↑ + 1 | Dominica | 30 | 27 | ↑ + 3 |
| Afghanistan | 31 | 31 | 0 | Cuba | 31 | 31 | 0 |

maintenance, and employee motivation [30]. Additionally, the World Health Organization (WHO) health framework evaluates HSP across four key areas: management, resource generation, service provision, and financing [5].

In our study, we critically examined various frameworks for evaluating HSP, aligning them with our research objectives. Given the significant impact of the COVID-19 pandemic on health systems, we separately evaluated its effects.

### 4.2. High-performing countries

Among the 31 countries assessed, Luxembourg emerges as the top performer. Its healthcare system ranks among the twelve most effective in Europe. Other high-performing systems include those in the Netherlands, Austria, Belgium, Denmark, Finland, France, Germany, Norway, Sweden, Switzerland, and the United Kingdom. These systems excel across various criteria, including access, health status, flexibility, innovation, quality, and resilience. Resilience, in particular, refers to a health system's ability to meet future population needs and adapt to innovations [31].

In Luxembourg, key institutions driving health system effectiveness include the Ministry of Health and the Social Security. These entities collaborate closely, make joint decisions, and assume responsibility for organizing, legislating, and financing the healthcare system [32]. Notably, Luxembourg boasts the second-highest health expenditure per capita

**Table 6. Ranking of 31 countries (Government Spending & GDP per capita Indices).**

| Government spending as % of total health spending | | | | GDP per capita | | | |
|---|---|---|---|---|---|---|---|
| Country | 2000 Rank | 2019 Rank | Change in rank | Country | 2000 Rank | 2019 Rank | Change in rank |
| Luxemburg | 1 | 2 | ↓ - 1 | Luxemburg | 1 | 1 | 0 |
| Cuba | 2 | 1 | ↑ + 1 | Switzerland | 2 | 2 | 0 |
| Germany | 3 | 4 | ↓ - 1 | USA | 3 | 3 | 0 |
| UK | 4 | 3 | ↑ + 1 | UK | 4 | 10 | ↓ - 6 |
| New Zealand | 5 | 5 | 0 | Qatar | 5 | 5 | 0 |
| Canada | 6 | 8 | ↓ - 2 | Canada | 6 | 8 | ↓ - 2 |
| Australia | 7 | 10 | ↓ - 3 | Singapore | 7 | 4 | ↑ + 3 |
| Dominica | 8 | 9 | ↓ - 1 | Germany | 8 | 7 | ↓ - 1 |
| Bahrain | 9 | 12 | ↓ - 3 | Australia | 9 | 6 | ↑ + 3 |
| Qatar | 10 | 6 | ↑ + 4 | New Zealand | 10 | 9 | ↑ + 1 |
| Bolivia | 11 | 7 | ↑ + 4 | Bahrain | 11 | 11 | 0 |
| Argentina | 12 | 13 | ↓ - 1 | Argentina | 12 | 15 | ↓ - 3 |
| Ukraine | 13 | 20 | ↓ - 7 | Libya | 13 | 16 | ↓ - 3 |
| Yemen | 14 | 25 | ↓ - 11 | Chile | 14 | 12 | ↑ + 2 |
| Libya | 15 | 11 | ↑ + 4 | Syria | 15 | 30 | ↓ - 15 |
| USA | 16 | 17 | ↓ - 1 | Dominica | 16 | 17 | ↓ - 1 |
| Brazil | 17 | 21 | ↓ - 4 | Brazil | 17 | 18 | ↓ - 1 |
| Paraguay | 18 | 18 | 0 | Cuba | 18 | 14 | ↑ + 4 |
| Iran | 19 | 22 | ↓ - 3 | Paraguay | 19 | 19 | 0 |
| Chile | 20 | 19 | ↑ + 1 | Iran | 20 | 23 | ↓ - 3 |
| Singapore | 21 | 15 | ↑ + 6 | Egypt | 21 | 21 | 0 |
| Haiti | 22 | 24 | ↓ - 2 | Bolivia | 22 | 22 | 0 |
| Syria | 23 | 14 | ↑ + 9 | China | 23 | 13 | ↑ + 10 |
| Egypt | 24 | 27 | ↓ - 3 | Haiti | 24 | 26 | ↓ - 2 |
| Kenya | 25 | 23 | ↑ + 2 | Ukraine | 25 | 20 | ↑ + 5 |
| Switzerland | 26 | 26 | 0 | Nigeria | 26 | 24 | ↑ + 2 |
| Uganda | 27 | 30 | ↓ - 3 | Yemen | 27 | 28 | ↓ - 1 |
| Niger | 28 | 28 | 0 | Kenya | 28 | 25 | ↑ + 3 |
| China | 29 | 16 | ↑ + 13 | Uganda | 29 | 27 | ↑ + 2 |
| Nigeria | 30 | 29 | ↑ + 1 | Niger | 30 | 29 | ↑ + 1 |
| Afghanistan | 31 | 31 | 0 | Afghanistan | 31 | 31 | 0 |

among European countries affiliated with the World Health Organization. Additionally, it holds the highest GDP per capita in Europe [31].

## 4.3. Countries with moderate performance

1. **Switzerland:**

   ◦ Switzerland ranks as the second-best country among the 31 countries in terms of health system performance (HSP).

   ◦ The Swiss healthcare system is widely recognized as one of the world's best. It operates under a mandatory health insurance (MHI) system, compulsory for all residents since 1996 [33].

   ◦ Despite its strengths, the Swiss system faces challenges due to its fragmented and federal structure, necessitating greater integration of care.

   ◦ While the system boasts high life expectancy and generally satisfies the population in terms of accessibility and quality of care, concerns persist regarding the representation of healthcare system users and the influence of patient lobbies.

**Table 7. Ranking of 31 countries (Deaths of COVID-19 & Vaccinations of COVID-19).**

| Deaths of COVID-19 | | | | Vaccinations of COVID-19 | | | |
|---|---|---|---|---|---|---|---|
| Country | 2020 Rank | 2023 Rank | Change in rank | Country | 2021 Rank | 2023 Rank | Change in rank |
| New Zealand | 1 | 9 | ↓ - 8 | China | 1 | 1 | 0 |
| Singapore | 2 | 7 | ↓ - 5 | USA | 2 | 2 | 0 |
| Dominica | 3 | 1 | ↑ + 2 | Brazil | 3 | 3 | 0 |
| Niger | 4 | 2 | ↑ + 2 | Germany | 4 | 4 | 0 |
| Cuba | 5 | 16 | ↓ - 11 | UK | 5 | 6 | ↓ - 1 |
| Haiti | 6 | 4 | ↑ + 2 | Iran | 6 | 5 | ↑ + 1 |
| Qatar | 7 | 3 | ↑ + 4 | Argentina | 7 | 8 | ↓ - 1 |
| Uganda | 8 | 12 | ↓ - 4 | Canada | 8 | 10 | ↓ - 2 |
| Bahrain | 9 | 6 | ↑ + 3 | Egypt | 9 | 9 | 0 |
| Luxembourg | 10 | 5 | ↑ + 5 | Chile | 10 | 12 | ↓ - 2 |
| Yemen | 11 | 8 | ↑ + 3 | Australia | 11 | 11 | 0 |
| Syria | 12 | 11 | ↑ + 1 | Cuba | 12 | 13 | ↓ - 1 |
| Australia | 13 | 19 | ↓ - 6 | Ukraine | 13 | 14 | ↓ - 1 |
| Nigeria | 14 | 10 | ↑ + 4 | Nigeria | 14 | 7 | ↑ + 7 |
| Libya | 15 | 14 | ↑ + 1 | Switzerland | 15 | 17 | ↓ - 2 |
| Kenya | 16 | 13 | ↑ + 3 | Singapore | 16 | 18 | ↓ - 2 |
| Afghanistan | 17 | 15 | ↑ + 2 | Bolivia | 17 | 19 | ↓ - 2 |
| Paraguay | 18 | 18 | 0 | Uganda | 18 | 15 | ↑ + 3 |
| China | 19 | 25 | ↓ - 6 | Kenya | 19 | 16 | ↑ + 3 |
| Switzerland | 20 | 17 | ↑ + 3 | New Zealand | 20 | 20 | 0 |
| Egypt | 21 | 21 | 0 | Paraguay | 21 | 21 | 0 |
| Bolivia | 22 | 20 | ↑ + 2 | Qatar | 22 | 22 | 0 |
| Canada | 23 | 22 | ↑ + 1 | Afghanistan | 23 | 27 | ↓ - 4 |
| Chile | 24 | 23 | ↑ + 1 | Bahrain | 24 | 26 | ↓ - 2 |
| Ukraine | 25 | 24 | ↑ + 1 | Libya | 25 | 25 | 0 |
| Argentina | 26 | 26 | 0 | Syria | 26 | 24 | ↑ + 2 |
| Germany | 27 | 28 | ↓ - 1 | Luxembourg | 27 | 28 | ↓ - 1 |
| Iran | 28 | 27 | ↓ - 1 | Niger | 28 | 23 | ↑ + 5 |
| UK | 29 | 29 | 0 | Yemen | 29 | 29 | 0 |
| Brazil | 30 | 30 | 0 | Haiti | 30 | 30 | 0 |
| USA | 31 | 31 | 0 | Dominica | 31 | 31 | 0 |

- ◦ Efforts to promote integrated care are viewed as a means to enhance HSP in Switzerland [34–37].

2. **Qatar:**

- ◦ Qatar holds the second rank among countries with moderate performance and also ranks third overall among the 31 countries.

- ◦ To address COVID-19 challenges, Qatar established a centralized digital platform through a community call center. Initially, it provided digital counseling via a hotline and later expanded to include a mass immunization COVID-19 vaccination hotline [38].

- ◦ Qatar has made significant strides in improving its healthcare system, investing heavily in infrastructure, research, and technology.

- ◦ Challenges remain, including the need for more medical workers in primary healthcare and addressing inequitable geographic distribution of healthcare resources [39,40].

## 4.4. Countries with low performance

In this study, we examine the performance of various countries, focusing on those that exhibit suboptimal outcomes. Among the nations analyzed, the United States, Australia, and Singapore secured the top three positions, ranking first, second, and third, respectively. Notably, they placed 4th, 5th, and 6th within the cohort of 31 countries.

The United States health system presents a complex landscape characterized by both strengths and vulnerabilities. On the positive side, it boasts a substantial and well-trained health workforce, a diverse array of high-quality medical specialists, and robust secondary and tertiary institutions. However, it grapples with several challenges. These include incomplete coverage for its citizenry, disproportionately high health expenditure per capita compared to other nations, subpar performance on objective and subjective quality measures, and an uneven distribution of resources and outcomes across different regions and population groups [41–43].

Turning our attention to Australia, its healthcare system centers around Medicare, which provides access to medical and hospital services for all Australian residents. Under this system, health professionals—including general practitioners, specialists, and optometrists—offer free or subsidized treatment. Additionally, public patients receive free care in public hospitals [44]. Despite the widespread availability of public services, public funding, and private health insurance [45], out-of-pocket expenditure (OOPE) still accounts for 20% of total healthcare spending across the OECD. This raises equity concerns, particularly for individuals with chronic conditions and greater healthcare needs, especially when those needs coincide with lower income [46].

At the other end of the spectrum, Afghanistan, Egypt, and Yemen occupy the lowest positions. Afghanistan, having endured two decades of war and the collapse of health institutions, embarked on healthcare system reforms, including primary healthcare, in 2002. However, the reality is starkly different from the perception that it relies solely on donor support. A significant portion of the country's total health expenditure is directly borne by households [47]. Notably, Afghanistan topped the list of out-of-pocket payments among the 31 countries in 2020, with approximately 77.58%. This ranking underscores the profound impact of war on the nation's healthcare system (WHO indicators, 2020).

Yemen, situated as one of the poorest countries in the Middle East, faces a prolonged conflict resulting from decades of political, economic, and social challenges. The ongoing war has severely affected public services, particularly the health sector, which now operates at less than half its capacity. Approximately 30.5 million Yemenis, of whom around 20.7 million require comprehensive humanitarian assistance, bear the brunt of this crisis. With over 80% of Yemen's population living below the poverty line, it stands as one of the most severe humanitarian crises globally [48]. Yemen's collapsing health system poses a grave threat to the well-being of its population [49].

Regarding out-of-pocket payments in 2020, Yemen ranks 30th out of 31 countries, with 63.96% (WHO indicators, 2020). The government's share in health payments remains small at 28.24 percent. High out-of-pocket expenses place a heavy burden on households, leading to catastrophic health expenditure [50]. Overall, Yemen occupies the last position in the ranking.

## 4.5. Health system performance in Iran

The study results indicate that Iran ranks 21st among the 31 countries under comparison. Furthermore, Iran falls into the category of countries with low performance, securing the 18th position within this group.

The program aimed at reforming Iran's healthcare system was initially devised before the 1979 revolution and subsequently implemented by health officials [51]. However, when benchmarked against Middle Eastern and Asian counterparts, Iran's performance in certain health indices remains unsatisfactory. For instance, the response rate in the Iranian health system stands at 67% for outpatient services and 73% for inpatient services—figures lower than those observed in Brazil and 14 European countries (where response rates exceed 81%) [52]. Additionally, approximately 70% of medical costs are borne by patients, annually pushing around 7% of the Iranian population below the poverty threshold [53].

The fourth five-year development plan highlights several challenges. These include an aging population, rising costs of care and health services, inadequate control of non-communicable diseases, increased prevalence of physical, mental, and social health conditions, and limited access to medical services in less developed regions. Furthermore, low satisfaction with paraclinical and non-essential services, coupled with escalating out-of-pocket payments, poses significant hurdles for national health officials [54].

The harsh economic sanctions against Iran constitute a major obstacle in implementing the Health System Reform Plan (HSRP). Although these sanctions ostensibly exclude food and medicines, limitations on financial transactions between Iran and the rest of the world, along with resource shortages due to the sanctions, have led to medicine scarcity within the Iranian health system. Consequently, the quality of domestically produced medicines used in medical services has declined [55].

Iran's unique geographic context, mass immigration from neighboring Afghanistan and Iraq, regional conflicts, insecurity, and illicit drug trafficking across its eastern borders with Pakistan and Afghanistan have collectively impacted the nation's health system [56].

Before the Islamic revolution, Iran witnessed over 5,000 maternal deaths annually, with a population of 35 million and an estimated 500,000 births per year. Presently, the absolute number of maternal deaths has decreased to less than 300 per year, despite ongoing challenges. Notably, Iran achieved Millennium Development Goal number 5 in 2008, seven years ahead of schedule [57]. In terms of maternal mortality rate in 2019, Iran ranked 16th among the 31 countries. Similarly, it held the 18th position for neonatal mortality rate within the same cohort (2019 data) [58].

Out-of-pocket (OOP) payments constitute over 50% of Iran's total healthcare costs, surpassing levels reported in other countries. This high OOP burden is closely associated with catastrophic health expenditure (CHE) and healthcare financing inequality. A study revealed that more than 5% of Iranian households experience CHE, with greater prevalence among those with lower socioeconomic status [59].

Over the past three decades, Iran has witnessed a significant increase in life expectancy. While its healthcare system has effectively controlled infectious diseases, it now grapples with the mounting burden of non-communicable diseases and injuries. Notably, Iran's health system has been more successful in reducing mortality rates than in managing complications and addressing mental health issues. This underscores the urgent need to expand rehabilitation services and integrate mental healthcare into primary care. Furthermore, Iran's healthcare system must enhance its preparedness and resilience to emerging health threats, as exemplified by the experience during the COVID-19 pandemic [60].

The evaluation of Iran's Health System Performance (HSP) and financing, in comparison with 30 selected countries, has revealed both strengths and weaknesses. While the country's health financing system exhibits relative stability, it remains fragile. Key factors, including politics, economics, and technology, significantly shape its structure [61].

However, this study faces limitations. Global Burden of Disease (GBD) studies have encountered criticism for various reasons. Doubts persist regarding the reliability and comprehensiveness of data sources, challenges in accurately quantifying certain health conditions, biases in data collection and modeling, and the intricate task of assessing the impact of environmental and social factors on health indicators. These criticisms have sparked debates about the accuracy and comparability of GBD estimates [62,63]. Another limitation pertains to data accessibility. Although numerous variables influence Health System Performance (HSP), our inability to access certain data prevented their inclusion in the country ranking.

Nevertheless, a notable strength of this study lies in its proposed framework and the comparison of countries across different economic and geographical levels. The examination of high-income countries such as the US and UK alongside low-income countries like Yemen and Nigeria provides valuable insights. Additionally, our focus on macro indicators related to health systems sheds light on crucial dimensions affecting overall performance.

## 5. Conclusion

The present study aimed to evaluate and rank Health System Performance (HSP). We selected 12 distinct variables from three levels: health, financial, and Covid-19 indicators. To determine the weight of each variable across the 31 surveyed countries, we employed a combination of three-scale weighting methods and entropy weighting methods. Subsequently, we ranked health system performance using the TOPSIS method and classified countries into three groups using hierarchical clustering.

The study results revealed substantial disparities in HSP. Only Luxembourg demonstrated strong performance, while Qatar and Switzerland achieved moderate rankings. The remaining countries fell into the group with poor performance. Generally, countries with favorable HSP were predominantly from Europe and North America. In the Middle East, only Qatar achieved an average performance, while South American and African countries also struggled with poor performance.

The variation in health system performance arises from multiple factors, including financing systems, health indicators, and economic and social development. Low-performing countries often grapple with insecurity and political instability, lacking sufficient financial resources to enhance health structures. Additionally, the rise in non-communicable diseases poses significant challenges. The Covid-19 pandemic further exacerbated these issues, imposing additional health and financial burdens on systems already strained by limited and unstable resources.

To enhance health system performance, countries must reduce patients' out-of-pocket payments and catastrophic health costs by bolstering sustainable financial resources. Developed nations, with greater financial capacity and strengthened health structures, serve as valuable models. Developing countries can learn from their experiences and focus on resource efficiency to achieve better economic and health outcomes.

## Supporting information

**S1 File. Health system Data.**
(ZIP)

## Author contributions

**Conceptualization:** Pirhossein Kolivand, Saeid Homayoun, Samad Azari.

**Data curation:** Peyman Saberian, Soheila Rajaie, Fereshte Karimi, Masoud Behzadifar.

**Formal analysis:** Arash Parvari.

**Investigation:** Pirhossein Kolivand, Samad Azari.

**Methodology:** Taher Dorooudi, Arash Parvari.

**Project administration:** Jalal Arabloo.

**Resources:** Peyman Saberian.

**Software:** Behzad Raei.

**Validation:** Pirhossein Kolivand, Fereshte Karimi.

**Writing – original draft:** Jalal Arabloo, Soheila Rajaie, Fereshte Karimi, Behzad Raei, Masoud Behzadifar, Arash Parvari, Peyman Namdar, Samad Azari.

**Writing – review & editing:** Jalal Arabloo, Taher Dorooudi, Seyed Jafar Ehsanzadeh, Shahrzad Salehbeigi, Saeid Homayoun, Samad Azari.

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
