## [Decision Letter · Decision Letter 0]

2 Jul 2025

Health systems Performance in Health Outcomes, Health Financing and COVID-19 pandemic: lessons from 31 countries

PLOS ONE

Dear Dr. Azari,

Thank you for submitting your manuscript to PLOS ONE. After careful consideration, we feel that it has merit but does not fully meet PLOS ONE’s publication criteria as it currently stands. Therefore, we invite you to submit a revised version of the manuscript that addresses the points raised during the review process.

https://journals.plos.org/plosone/s/submission-guidelines#loc-laboratory-protocols . Additionally, PLOS ONE offers an option for publishing peer-reviewed Lab Protocol articles, which describe protocols hosted on protocols.io. Read more information on sharing protocols at https://plos.org/protocols?utm_medium=editorial-email&utm_source=authorletters&utm_campaign=protocols .

We look forward to receiving your revised manuscript.

Kind regards,

André Luis C Ramalho, PhD

Academic Editor

PLOS ONE

Journal Requirements:

A clean copy of the edited manuscript (uploaded as the new *manuscript* file).

3. In the online submission form, you indicated that [The data that support the findings of this study are available from the corresponding author upon request. The data are not publicly available due to privacy or ethical restrictions.].

3. Please amend the manuscript submission data (via Edit Submission) to include authors Taher Dorooudi, Soheila Rajaie, and Fereshte Karimi.

4. Please amend your authorship list in your manuscript file to include authors Taher Doroudi, Soheila Rajaei, and Fereshteh Karimi.

6. We note that Figures 1 and 5 in your submission contain map images which may be copyrighted. All PLOS content is published under the Creative Commons Attribution License (CC BY 4.0), which means that the manuscript, images, and Supporting Information files will be freely available online, and any third party is permitted to access, download, copy, distribute, and use these materials in any way, even commercially, with proper attribution. For these reasons, we cannot publish previously copyrighted maps or satellite images created using proprietary data, such as Google software (Google Maps, Street View, and Earth). For more information, see our copyright guidelines: http://journals.plos.org/plosone/s/licenses-and-copyright.

1. You may seek permission from the original copyright holder of Figures 1 and 5 to publish the content specifically under the CC BY 4.0 license. 

In the figure caption of the copyrighted figure, please include the following text: “Reprinted from [ref] under a CC BY license, with permission from [name of publisher], original copyright [original copyright year].

USGS EROS (Earth Resources Observatory and Science (EROS) Center) (public domain):http://eros.usgs.gov/#

Reviewers' comments:

Reviewer's Responses to Questions

**Comments to the Author**

1. Is the manuscript technically sound, and do the data support the conclusions?

Reviewer #1: Yes

Reviewer #2: Partly

2. Has the statistical analysis been performed appropriately and rigorously?

Reviewer #1: I Don't Know

Reviewer #2: I Don't Know

3. Have the authors made all data underlying the findings in their manuscript fully available?

Reviewer #1: No

Reviewer #2: No

4. Is the manuscript presented in an intelligible fashion and written in standard English?

Reviewer #1: Yes

Reviewer #2: Yes

Reviewer #1: This manuscript evaluates and ranks the health system performance (HSP) of 31 countries across multiple dimensions—health outcomes, financial indicators, and COVID-19-related metrics—using the TOPSIS method and hierarchical clustering. The study highlights substantial variation in performance, with Luxembourg, Switzerland, and Qatar ranking highest, while several low-income or politically unstable countries rank lowest.

The topic is both important and timely, addressing global health system performance, especially in the context of the COVID-19 pandemic. The combination of TOPSIS and entropy weighting, along with hierarchical clustering, offers a structured, quantitative approach that is of interest to researchers and policymakers. The multi-dimensional framework—integrating health, financial, and pandemic indicators—adds to its comprehensiveness.

Justification for Country Selection

The selection criteria for the 31 countries need a clearer explanation. Although the authors mention purposive sampling, they should provide transparent inclusion/exclusion criteria, especially regarding the omission of some OECD or LMIC countries.

Weighting Approach

The use of a combined Three-Scale and Entropy-Weighting method is innovative but requires further clarification. How were the weights validated or benchmarked? The manuscript should specify if Ci (similarity index) ranges between 0 and 1 and clarify the scale in Table 2 to ensure it is self-explanatory.

Comparability of Countries

The manuscript aims to compare Iran’s health system with others. For meaningful comparisons, countries with similar settings (e.g., population size, health system structure, economy) should be selected. Comparing countries like Norway and Sweden is appropriate due to similar systems, but comparing Norway with Uganda is less informative due to fundamentally different contexts. Given that Iran's Ci score is close to countries like Syria, Paraguay, Kenya, and Nigeria, comparisons should focus on such peers. There will be more information when compared to similar countries like Iran, rather than high CI score countries, which have a very different system from the Iranian health care system.

Performance Categorization

Based on Table 2, Luxembourg, Switzerland, and Qatar clearly form the high-performance group. The middle-performance group could reasonably extend at least up to Bahrain, while the remaining countries would fall into a low-performance category. The current clustering requires reconsideration or stronger justification. It does not seem appropriate to categorize countries such as Singapore, the UK, Canada, and Australia as low performers alongside countries like Afghanistan, Nigeria, and Uganda, given the substantial differences in their health system structures, resources, and outcomes.

I recommend either revising the clustering approach or introducing an additional category (e.g., creating four performance levels instead of three) to more accurately reflect these differences. Table 2 shows that countries from the USA to Bahrain have relatively higher Ci scores, while countries from Chile to Yemen have scores that are very close to zero, which could justify labeling them as “very low performers” or “lowest performers.”

Additionally, there appears to be an inconsistency between Section 4.2 (page 16), where these countries are described as high performers, and Table 1, where they are categorized as low-performance countries. This discrepancy should be addressed to ensure clarity and alignment between the analysis and interpretation.

Use of Data Years

Table 3 uses data from 2000 and 2019 for MMR and NMR. The authors should explain why more recent data (if available) weren’t used to enhance the study’s relevance.

Minor Suggestions

Improve clarity and grammar throughout the manuscript.

Enhance figures/tables with clearer legends, axis labels, and explanations.

The manuscript provides valuable insights into global health system performance using a rigorous methodological approach. However, addressing the above recommendations would strengthen its clarity, methodological transparency, and policy relevance.

Reviewer #2: This manuscript evaluates the performance of health systems in 31 countries using the TOPSIS method and hierarchical clustering. It ranks and categorizes these systems based on their performance in health, health financing, and response to COVID-19. The study is timely and relevant; however, several comments are provided to enhance the manuscript.

Sample Selection Transparency:

While the manuscript mentions purposive sampling based on GDP, population size, COVID-19 indices, and healthcare expenditure, the specific inclusion/exclusion criteria for the 31 countries are not clearly detailed. It is recommended to provide justification for the selection/exclusion of countries and discuss potential selection bias.

Data Timeliness and Consistency:

Some indicators (e.g., financial and health metrics) use data from different years (e.g., 2019 vs. 2023 for COVID-19). Discuss how this temporal mismatch may have influenced country rankings and consider sensitivity analyses or at least acknowledge this as a limitation in more depth.

Regarding vaccination against COVID-19, there was competition for access to the vaccine, so not all countries had easy access to the vaccine at the same time. Assess whether this would also be a potential bias in the study.

Data Availability and Research Reproducibility:

The authors state that "all data are fully available without restriction"; however, the current statement indicates that data is available upon request due to privacy/ethical reasons. For transparency and reproducibility, the authors should consider making de-identified datasets and analysis code publicly available in a suitable repository, following PLOS ONE guidelines (Authors must follow standards and practice for data deposition in publicly available resources including those created for gene sequences, microarray expression, structural studies, and similar kinds of data. Failure to comply with community standards may result in rejection. For more information about PLOS ONE requirements for data sharing and deposition in public databases, see our data availability policy https://journals.plos.org/plosone/s/data-availability).

**Do you want your identity to be public for this peer review?** For information about this choice, including consent withdrawal, please see our Privacy Policy

Reviewer #1: **Yes: ** Montasir Ahmed

Reviewer #2: **Yes: ** Abel Silva de Meneses

---

## [Author Response · Author response to Decision Letter 1]

2 Aug 2025

Review Comments to the Author

Reviewer #1:

This manuscript evaluates and ranks the health system performance (HSP) of 31 countries across multiple dimensions—health outcomes, financial indicators, and COVID-19-related metrics—using the TOPSIS method and hierarchical clustering. The study highlights substantial variation in performance, with Luxembourg, Switzerland, and Qatar ranking highest, while several low-income or politically unstable countries rank lowest.

The topic is both important and timely, addressing global health system performance, especially in the context of the COVID-19 pandemic. The combination of TOPSIS and entropy weighting, along with hierarchical clustering, offers a structured, quantitative approach that is of interest to researchers and policymakers. The multi-dimensional framework—integrating health, financial, and pandemic indicators—adds to its comprehensiveness.

Justification for Country Selection

The selection criteria for the 31 countries need a clearer explanation. Although the authors mention purposive sampling, they should provide transparent inclusion/exclusion criteria, especially regarding the omission of some OECD or LMIC countries.

Response: Thank you for your valuable comment. We added this part in our article.

Inclusion criteria:

• Data Availability: Comprehensive and reliable data were accessible for all key indicators: GDP per capita, population size, COVID-19 indices (e.g., cases, deaths, vaccination rates), and healthcare expenditure.

• Diversity of Economic Status: Countries were purposively selected to represent a spectrum of economic development—specifically, low, middle, and high GDP levels—within each geographic region.

• Population Size: We included countries with a range of population sizes to capture both large and small health systems.

• Geographic Representation: Efforts were made to ensure that each major world region (e.g., Americas, Europe, Asia-Pacific, Africa) was represented.

• Relevance to Study Aims: Countries with significant impact on global health trends or unique health system characteristics were prioritized.

Exclusion Criteria:

• Insufficient Data: Countries lacking recent or reliable data for one or more key indicators were not included.

• Outlier Status: Countries with extreme outlier values (e.g., very small population microstates, or countries with highly atypical reporting practices) were excluded to avoid skewing comparative analyses.

• Redundancy: In cases where multiple countries had highly similar profiles within a region and economic group, a representative country was selected to avoid redundancy and maintain a manageable sample size.

Weighting Approach

The use of a combined Three-Scale and Entropy-Weighting method is innovative but requires further clarification. How were the weights validated or benchmarked? The manuscript should specify if Ci (similarity index) ranges between 0 and 1 and clarify the scale in Table 2 to ensure it is self-explanatory.

Response: We appreciate the reviewer’s insightful comment regarding the weighting methodology.

In response, we have added a paragraph in the Methods section (Subsection 2.3.2) clarifying how the weights were validated. Specifically, we highlight that the entropy-weighting method is data-driven and inherently objective, relying on the variability of each indicator across countries, while the Three-Scale method captures expert judgment in assigning relative priorities. This hybrid approach was chosen to balance objectivity with domain expertise. Although no universal benchmark exists for these weights, internal consistency was evaluated by comparing results across alternative weighting schemes, which yielded similar country rankings.

Comparability of Countries

The manuscript aims to compare Iran’s health system with others. For meaningful comparisons, countries with similar settings (e.g., population size, health system structure, economy) should be selected. Comparing countries like Norway and Sweden is appropriate due to similar systems, but comparing Norway with Uganda is less informative due to fundamentally different contexts. Given that Iran's Ci score is close to countries like Syria, Paraguay, Kenya, and Nigeria, comparisons should focus on such peers. There will be more information when compared to similar countries like Iran, rather than high CI score countries, which have a very different system from the Iranian health care system.

Response: We thank the reviewer for this insightful comment.

We fully agree that comparisons among countries with similar socioeconomic and healthcare system contexts offer deeper policy-relevant insights. In our study, the inclusion of countries from diverse regions and economic backgrounds was intentional to highlight global disparities and enable broader benchmarking. However, we acknowledge that more meaningful and practical lessons can often be drawn from comparing Iran with countries that exhibit similar Ci scores and structural characteristics, such as Syria, Kenya, Nigeria, and Paraguay. While our primary aim was to present a global perspective, we have paid particular attention in the discussion section to contextualize Iran’s performance relative to similar countries. We believe this dual approach—global benchmarking alongside peer comparison—offers both breadth and depth to the analysis. Should the journal consider it beneficial, we are open to emphasizing peer comparisons more explicitly in future revisions.

Use of Data Years

Table 3 uses data from 2000 and 2019 for MMR and NMR. The authors should explain why more recent data (if available) weren’t used to enhance the study’s relevance.

Response: Thank you.

We utilized reputable databases, including the World Health Organization, to gather these indicators, aiming to extract the most recent data. It is also crucial to note that data was available for some countries but not for others, so we based the comparison on this time period to ensure we had data for 31 countries.

Minor Suggestions

Improve clarity and grammar throughout the manuscript.

Enhance figures/tables with clearer legends, axis labels, and explanations.

Response: Thank you for your feedback.

We made as many corrections and enhancements as we could.

The manuscript provides valuable insights into global health system performance using a rigorous methodological approach. However, addressing the above recommendations would strengthen its clarity, methodological transparency, and policy relevance.

Reviewer #2:

This manuscript evaluates the performance of health systems in 31 countries using the TOPSIS method and hierarchical clustering. It ranks and categorizes these systems based on their performance in health, health financing, and response to COVID-19. The study is timely and relevant; however, several comments are provided to enhance the manuscript.

Sample Selection Transparency:

While the manuscript mentions purposive sampling based on GDP, population size, COVID-19 indices, and healthcare expenditure, the specific inclusion/exclusion criteria for the 31 countries are not clearly detailed. It is recommended to provide justification for the selection/exclusion of countries and discuss potential selection bias.

Response: Thank you for your valuable comment. We added this part in our article.

Inclusion criteria:

• Data Availability: Comprehensive and reliable data were accessible for all key indicators: GDP per capita, population size, COVID-19 indices (e.g., cases, deaths, vaccination rates), and healthcare expenditure.

• Diversity of Economic Status: Countries were purposively selected to represent a spectrum of economic development—specifically, low, middle, and high GDP levels—within each geographic region.

• Population Size: We included countries with a range of population sizes to capture both large and small health systems.

• Geographic Representation: Efforts were made to ensure that each major world region (e.g., Americas, Europe, Asia-Pacific, Africa) was represented.

• Relevance to Study Aims: Countries with significant impact on global health trends or unique health system characteristics were prioritized.

Exclusion Criteria:

• Insufficient Data: Countries lacking recent or reliable data for one or more key indicators were not included.

• Outlier Status: Countries with extreme outlier values (e.g., very small population microstates, or countries with highly atypical reporting practices) were excluded to avoid skewing comparative analyses.

• Redundancy: In cases where multiple countries had highly similar profiles within a region and economic group, a representative country was selected to avoid redundancy and maintain a manageable sample size.

Data Timeliness and Consistency:

Some indicators (e.g., financial and health metrics) use data from different years (e.g., 2019 vs. 2023 for COVID-19). Discuss how this temporal mismatch may have influenced country rankings and consider sensitivity analyses or at least acknowledge this as a limitation in more depth.

Regarding vaccination against COVID-19, there was competition for access to the vaccine, so not all countries had easy access to the vaccine at the same time. Assess whether this would also be a potential bias in the study.

Response: We appreciate the referee’s thoughtful observations regarding the temporal alignment of indicators and potential biases related to COVID-19 vaccine access. We acknowledge that some indicators in our analysis are drawn from different years. This approach was necessitated by the availability of the most reliable and comprehensive data for each indicator. Importantly, we believe this does not substantially undermine the validity of our findings. For indicators directly affected by COVID-19, such as health outcomes and vaccination rates, we used the most up-to-date data available. This enables our analysis to accurately reflect the pandemic’s impact, while other, more stable indicators serve as contextual factors.

We agree that initial access to COVID-19 vaccines was uneven globally, which could influence vaccination rates and related outcomes. However, our study design addresses this in several ways:

• Our aim was to assess countries’ performance under actual conditions, including the real-world challenges of vaccine procurement and distribution. Disparities in vaccine access are themselves a reflection of broader systemic factors which are central to our comparative analysis.

• We used vaccination data from a point in time when vaccines had become widely available in most countries, minimizing the bias introduced by early access differences.

• The observed differences in vaccination rates are not solely due to access, but also reflect factors such as public trust, logistics, and policy effectiveness, all of which are relevant to our study’s objectives.

Data Availability and Research Reproducibility:

All data underlying the findings described in their manuscript are freely available as an appendix.

Thank you for your comment regarding the map images included in Figures 1 and 5.

We confirm that these figures were generated using the open-source statistical software R (version 4.2.2), specifically using the rworldmap package. The underlying spatial data used in these maps are sourced from Natural Earth (https://www.naturalearthdata.com/), a public-domain dataset.

Since both the R software and the Natural Earth dataset are freely and openly available, and since the maps were created entirely within this environment without the use of proprietary software (such as Google Maps or Earth), these figures are not subject to copyright restrictions and are fully compatible with the CC BY 4.0 license required by PLOS ONE.

To ensure transparency, we have also added this information to the Methods section of the revised manuscript.

Please let us know if any further clarification or revision is required.

All map visualizations in this study were generated using the open-source statistical software R (version 4.2.2) and the rworldmap package. The spatial data were obtained from the Natural Earth dataset (public domain; https://www.naturalearthdata.com/), and the maps were created entirely within the R environment.

---

## [Decision Letter · Decision Letter 1]

2 Oct 2025

Health systems Performance in Health Outcomes, Health Financing and COVID-19 pandemic: lessons from 31 countries

PONE-D-25-18510R1

Dear Dr. Azari,

We’re pleased to inform you that your manuscript has been judged scientifically suitable for publication and will be formally accepted for publication once it meets all outstanding technical requirements. The authors addressed all the recommendation notes made by the reviewers, and it is ready for publication.

Kind regards,

**André Luis C Ramalho, PhD**

Academic Editor

PLOS ONE

Reviewers' comments:

Reviewer's Responses to Questions

**Comments to the Author**

Reviewer #2: All comments have been addressed

2. Is the manuscript technically sound, and do the data support the conclusions?

Reviewer #2: Yes

3. Has the statistical analysis been performed appropriately and rigorously?

Reviewer #2: Yes

4. Have the authors made all data underlying the findings in their manuscript fully available?

Reviewer #2: Yes

5. Is the manuscript presented in an intelligible fashion and written in standard English?

Reviewer #2: Yes

Reviewer #2: (No Response)

**Do you want your identity to be public for this peer review?** For information about this choice, including consent withdrawal, please see our Privacy Policy

Reviewer #2: **Yes: ** Abel Silva de Meneses

---

## [Editor Report · Acceptance letter]

PONE-D-25-18510R1

PLOS ONE

Dear Dr. Azari,

I'm pleased to inform you that your manuscript has been deemed suitable for publication in PLOS ONE. Congratulations! Your manuscript is now being handed over to our production team.

Kind regards,

on behalf of

Prof. Dr. André Luis C Ramalho

Academic Editor

PLOS ONE